# Hemorrhage and Sudden Death in a Cat with Pancreatic Hemangiosarcoma

**DOI:** 10.3390/vetsci10010008

**Published:** 2022-12-24

**Authors:** Corina Toma, Roxana Popa, Mara-Georgiana Haralambie, Oana-Roxana Haralambie, Raluca Marica

**Affiliations:** Department of Veterinary Pathology, Faculty of Veterinary Medicine, University of Agricultural Sciences and Veterinary Medicine, 400372 Cluj-Napoca, Romania

**Keywords:** cancer, feline, hemangiosarcoma, pancreas

## Abstract

**Simple Summary:**

Feline pancreatic neoplasia is a rare clinical and post-mortem diagnosis. This report aims to present the pathological findings of a case of pancreatic hemangiosarcoma in a cat with a history of sudden death. A neoplastic mass, covered by numerous small blood clots, was identified within the body of the pancreas, along with multiple metastases within the omentum and liver. The sudden death was most likely caused by the spontaneous rupture of the pancreatic neoplasm, which produced severe hemoperitoneum and anemia. Cytological and histological findings were completed by immunohistochemical evaluation, supporting the diagnosis of hemangiosarcoma.

**Abstract:**

A 15-year-old female, indoor, spayed, domestic shorthair cat was presented for post-mortem evaluation with a history of sudden death. A red-brown 3 cm x 4 cm neoplastic mass was identified within the body of the pancreas, along with multiple similar nodular structures within the omentum and liver. Associated lesions included hemoperitoneum, yellow discoloration of the peritoneal wall, and severe anemia. Spindle-shaped neoplastic cells exhibiting malignancy features, which occasionally contained within the cytoplasm haematoidin crystals and/or red blood cells, were observed during cytological examination. Histologically, the neoplastic cells were organized in short streams forming vascular spaces filled with erythrocytes. Immunohistochemically, the neoplastic cells were immunolabeled for vimentin and CD31, supporting the diagnosis of hemangiosarcoma. This report offers a complete post-mortem and histological evaluation of a rare tumor in cats with an unusual location and a comparative assessment of 3 anti-CD31 antibodies.

## 1. Introduction

Hemangiosarcoma (HSA) is an aggressive malignant tumor originating in the vascular endothelial cells. In both dogs and cats, HSA is classified into non-visceral and visceral forms [1,2], the latter being more commonly diagnosed in cats [3]. Visceral HSA accounts for 0.04% of all neoplasms in felines, and the most common locations are the liver, small intestine, large intestine, lymph node, omentum, spleen, and lung [3]. In most cases (77%), at the time of diagnosis, visceral HSA are found along with multiple metastases [4]. The incidence of pancreatic tumors in cats accounts for less than 0.5% [5], and primary pancreatic HSA was sporadically reported [3,6,7]. Although there are few studies concerning the prognosis of HSA, both visceral and non-visceral forms appear to have a poor prognosis [3,8]. There is no recorded breed or sex predilection, and usually, it is diagnosed in middle-aged to older cats [8]. Clinical signs may vary according to the location of the primary tumor and the presence of metastasis, but usually, they include lethargy, anorexia, dyspnea, tachypnea, emesis, abdominal pain, and hypothermia [3,9]. 

In cats, sudden death was recorded in 7.9% of cases, and the most common cause of unexpected death was related to trauma, followed by heart disease and neoplastic disease [10]. 

Currently, there are only a few reports in the literature on the subject of feline primary and metastatic non-epithelial pancreatic tumors, most of them being included in epidemiological studies [3,6,7,11]. The aim of this case report was to present in detail the post-mortem findings and the cytological, histological, and immunohistochemical diagnosis of a pancreatic HSA with multiple metastases in a cat with a history of sudden death. 

## 2. Materials and Methods

A 15-year-old female, indoor, spayed, domestic shorthair cat with a history of sudden death was presented to the Department of Pathology at the Faculty of Veterinary Medicine (Cluj-Napoca) for post-mortem evaluation. According to the owner, the cat had no clinical signs before death. A gross general pathological examination was performed, and the body score was defined as good.

Tissue samples collected during the post-mortem evaluation consisted of pancreatic neoplasm and multiple abdominal metastases. Impression smears were obtained from all nodular masses and stained with Dia Quick Panoptic (DQP, Reagents Kft, Budapest, Hungary).

Neoplastic masses from each affected organ were collected and fixed for 48–72 h in 10% neutral buffered formalin, followed by routine paraffin embedding. Two-three μm thick tissue sections were obtained using a manual microtome (Thermo Scientific^TM^ HM 325 Rotary Microtome), followed by hematoxylin & eosin staining.

For immunohistochemical evaluation, the panel of antibodies is presented in Table 1. For each antibody listed, there were also negative controls used. The reactions were carried out using the Leica Bondmax^TM^ Immunohistochemistry system (Bond Max model [M2 12154 series], Leica Biosystems, Melbourne, Australia). 

All histological sections were examined using an Olympus BX-41 microscope. The photomicrographs were obtained using Olympus SP 350 digital camera connected to Stream Basic imaging software (Olympus Corporation, Tokyo, Japan). 

## 3. Results

During the external evaluation, the most important changes were pale mucous membranes and moderate abdominal distention. Approximately 300 mL of partially coagulated blood was detected within the abdominal cavity, associated with yellow discoloration of the peritoneal wall and abdominal fat. A mottled red-brown 3 cm × 4 cm neoplastic mass was found (Figure 1A), located in the body of the pancreas, adjacent to the duodenum (Figure 1B). The surface of the neoplastic mass was covered by numerous small blood clots. The neoplasm showed an infiltrative growth pattern, partially surrounding the duodenum and a soft-friable consistency. Focally, the omentum was finely attached to the neoplastic mass, and multiple 1–4 mm red-brown nodular structures (Figure 1C) were present on the omental surface. Diffusely, the liver showed a yellow discoloration, and within the hepatic parenchyma, multiple nodular masses (Figure 1D), ranging from 0.3 to 2 cm in diameter, with a similar appearance to the pancreatic neoplasm, were observed.

Additional findings included moderate pulmonary edema, chronic, mild interstitial nephritis, a renal cyst, and follicular gastritis.

Cytological examination revealed the presence of neoplastic spindle cells (Figure 2A), approximately 30–35 × 10–15 µm in size, mostly arranged individually. Most of the neoplastic cells contained intracytoplasmic well-demarcated, clear vacuoles and, rarely, either intact erythrocytes or hematoidin crystals. The nuclei were oval to elongated in shape, centrally/paracentrally located, and containing up to 3 nucleoli. Macronucleoli were frequently found and rarely were mitotic figures and binucleated cells observed. The nuclear-cytoplasmic ratio (N/C) was moderate to high.

Histologically, the pancreatic parenchyma was focally extensive and replaced by a non-encapsulated, poorly demarcated, infiltrative neoplasm (Figure 2B), composed of the spindle to polygonal cells, organized in streams and bundles, delineating slit-like, variably-sized vascular spaces (Figure 2C). Neoplastic cells showed variably evident cellular borders, a moderate amount of finely fibrillar, occasionally vacuolated cytoplasm, and a moderate N/C ratio. Throughout the neoplasm, multiple variably sized areas of hemorrhage were noticed, as well as extensive areas of necrosis. The number of mitotic figures ranged between 2 and 6/ 2.37 mm^2^, and atypical cell divisions were a frequent finding. Similar histologic features were observed in the hepatic and omental nodular structures. The established presumptive diagnosis was pancreatic hemangiosarcoma with multiple metastases based on gross, cytological, and histological findings. Immunohistochemically, the neoplastic cells showed diffuse cytoplasmic staining for vimentin (Figure 2D). At the same time, there was no reaction recorded for multi-cytokeratin (Figure 2E) and von Willebrand factor (Figure 2F), but for the latter, there was no reactivity with this species. CD31, clone JC70A showed the best results: strong membranous immunoexpression and no background staining (Figure 2G), confirming the endothelial origin of the neoplastic cells. Using the 1A10 clone (Figure 2H), the intensity of immunoexpression was very weak by comparison with the JC70A clone. At the same time, the polyclonal CD31 (Figure 2I) showed no reactivity with this species as the immunolabelling was absent in the internal control (non-neoplastic endothelial cells). 

## 4. Discussion

Although with a low prevalence (0.5%), the most common pancreatic neoplasms in cats are represented by epithelial tumors, adenocarcinoma is the most frequently identified one [5,12]. Non-neoplastic lesions were also reported in cats; for instance: Acinar cell hyperplasia has been identified as a common incidental finding in middle-aged to older individuals. Ductular hyperplasia and pancreatic cysts are also included in this group but were sporadically reported in this species [13]. Another incidental non-neoplastic finding described as a pancreatic lesion is the ectopic spleen, which may be occasionally found in cats [14].

Non-epithelial neoplasms of the pancreas, such as lymphoma, mast cell tumor, fibrosarcoma, histiocytic sarcoma, and hemangiosarcoma, are rarely encountered and poorly characterized [11]. To the authors’ knowledge, there are only six cases of primary pancreatic HSA described in the literature (Table 2), none of them specifically reporting the presence of metastases or detailing the pathological findings [3,6,7,11]. 

According to the clinical history provided by the owner, the cat presented no clinical signs before death. Considering the large amount of partially coagulated blood found within the peritoneal cavity, the time elapsed between the onset of hemoperitoneum and the death of the cat was very short. Post-mortem findings suggest that spontaneous rupture of the pancreatic neoplasm, producing hemoperitoneum and severe anemia, was responsible for the sudden death. This evidence was supported by the blood clots observed on the surface of the neoplastic mass. Hemoperitoneum may be classified into spontaneous and traumatic [15], the latter being the most frequent cause of hemoperitoneum in cats [6].

Even though the main metastasizing route of HSA is represented by the hematogenous one, the friable nature of this neoplasm may cause small tears making it prone to direct implantation [13]. 

Histological evaluation of the pancreatic mass, omental and hepatic metastases revealed the presence of neoplastic spindle to polygonal cells organized in streams and bundles, which were delineating variably-sized vascular spaces. Recognized growth patterns of HSA include capillary, cavernous, solid, and epithelioid variants [16]. Currently, there is no accepted grading scheme for HSA in either dogs or cats, but the prognosis is considered poor in both species, regardless of the location of the primary neoplasm [17]. 

For immunohistochemical evaluation, the markers used for diagnosis of HSA are CD31 and/or von Willebrand factor. Additionally, CD34 can be used if the immunostaining for the previously mentioned markers results in a strong background. CD34 expression was also observed in several types of carcinomas and lymphohematopoietic tumors in cats, and its specificity for HSA is considered lower than the other two markers [18]. Using the antibody CD31, clone JC70A, a diffuse and strong membranous immunoexpression was observed, supporting the diagnosis of HSA. According to previous studies, clone JC70A showed similar results [18]. The immunolabelling was present in both neoplastic and non-neoplastic endothelial cells. In contrast, von Willebrand factor, clone 36B11, showed no reactivity with this species, as there was a lack of immunolabelling of normal endothelial cells. In dogs, recent studies have shown weak immunolabeling or even negative results in hemangiosarcomas with a low degree of differentiation for both CD31 and von Willebrand factor antibodies [13,19], but there are no similar studies in cats. 

## 5. Conclusions

In both dogs and cats, HSA is considered an aggressive tumor with a poor prognosis and high potential for metastasis. This report describes the pathological findings of a case of pancreatic HSA with abdominal metastasis and hemoperitoneum. The gross and histological findings were supported by the immunohistochemical evaluation, with the best results being obtained using CD31, clone JC70A. HSA should be considered as a differential diagnosis of pancreatic masses in cats.

## Figures and Tables

**Figure 1 vetsci-10-00008-f001:**
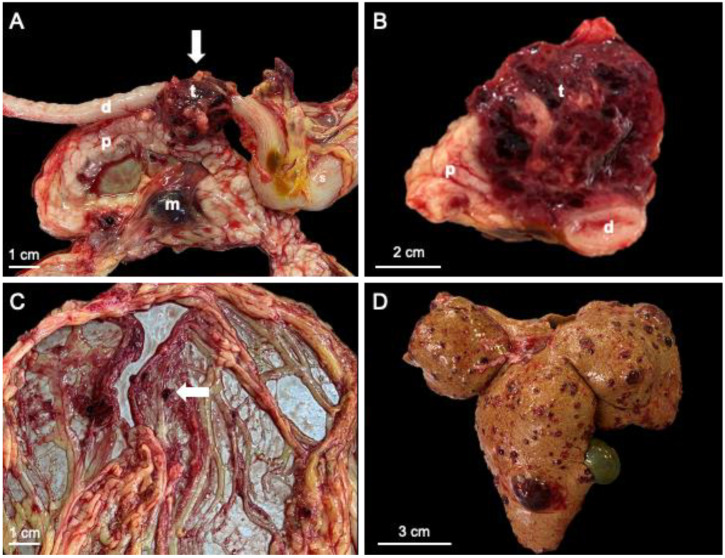
Gross findings. (**A**) The neoplastic mass (arrow) was located within the body of the pancreas, adjacent to the duodenum; (**B**) On cross-section, the neoplasm showed a mottled appearance and an infiltrative growth pattern, partially surrounding the duodenum; (**C**) Multiple metastases (arrow) were identified within the omentum; (**D**) The liver showed a diffuse pale discoloration and multiple variably-sized metastatic nodules; d—duodenum; m—metastasis; p—pancreas; s—stomach; t—tumor.

**Figure 2 vetsci-10-00008-f002:**
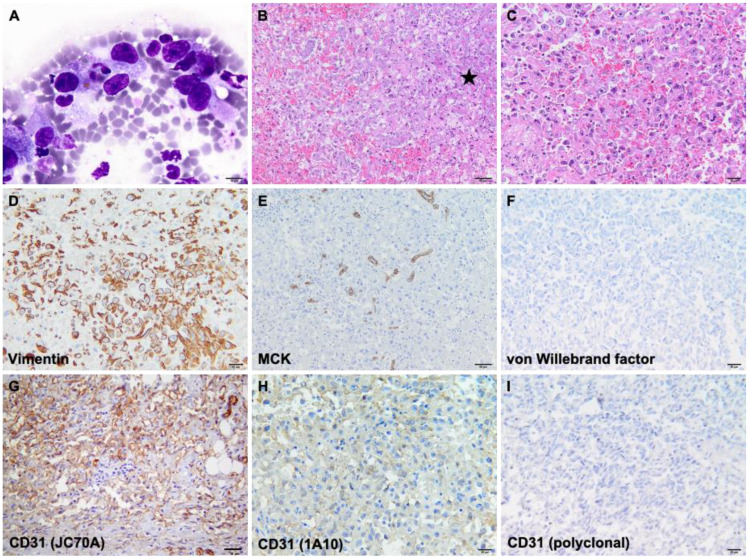
Cytological, histological, and immunohistochemical features of pancreatic HSA. (**A**) Spindle-shaped neoplastic cells with moderate anisokaryiosis and anisocytosis that occasionally contained within the cytoplasm hematoidin crystals and/or red blood cells; (**B**) Solid and capillary areas of HSA replaced the pancreatic parenchyma (asterisk); (**C**) Epithelioid area of HSA, characterized by polygonal neoplastic cells separated by red blood cells; (**D**) The neoplastic cells showed diffuse cytoplasmic immunoexpression of vimentin; (**E**) The remaining acinar pancreatic cells were visible as multi-cytokeratin-positive cells at the periphery of the neoplastic mass; (**F**) von Willebrand factor showed no reactivity; (**G**) CD31, clone JC70A revealed diffuse membranous immunolabelling and no background staining; (**H**) CD31, clone 1A10 showed membranous immunolabelling of the neoplastic cells and moderate background staining; (**I**) Polyclonal CD31 showed no reactivity.

**Table 1 vetsci-10-00008-t001:** Primary antibodies used for the diagnosis of HSA.

Antibody	Clone	Company	Positive Control
Mouse anti-human monoclonal to vimentin	SRL33	Leica Biosystems Newcastle Ltd., Newcastle upon Tyne, UK	Endothelial cells
Mouse anti-human to multi-cytokeratin	AE1/AE3	Leica Biosystems Newcastle Ltd., Newcastle upon Tyne, UK	Pancreatic acinar cells
Rabbit polyclonal to CD31	polyclonal	Abcam, Cambridge, UK	Endothelial cells
Mouse anti-human monoclonal to CD31	JC70A	Leica Biosystems Newcastle Ltd., Newcastle upon Tyne, UK	Endothelial cells
Mouse anti-human monoclonal to CD31	1A10	Leica Biosystems Newcastle Ltd., Newcastle upon Tyne, UK	Endothelial cells
Mouse anti-human monoclonal to von Willebrand factor	36B11	Leica Biosystems Newcastle Ltd., Newcastle upon Tyne, UK	Endothelial cells

**Table 2 vetsci-10-00008-t002:** Published cases of pancreatic HSA in cats.

Case	Source	Clinical findings
1	Engle and Brodey, 1969 [6]	Not reported
2	Ottenjann et al., 2003 [7]	Not reported
3	Culp et al., 2008 [3]	Not reported
4	Culp et al., 2008 [3]	Not reported
5	Törner et al., 2020 [11]	Palpable abdominal mass
6	Törner et al., 2020 [11]	Not reported

## Data Availability

Not applicable.

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
