# Peer review of "Hemorrhage and Sudden Death in a Cat with Pancreatic Hemangiosarcoma"

_vetsci, 2022, doi:10.3390/vetsci10010008_

Round 1

Reviewer 1 Report

Interesting and curious work that contributes to the knowledge about neoplasms in domestic cats.

Well written article with excellent macroscopic and microscopic images.

Author Response

Dear Reviewer,

Thank you very much for reviewing this paper and for the positive feedback.

Reviewer 2 Report

Dear authors,

you present a well written and concise case report. However, the tumor entity (hemangiosarcoma) has been reported in the affected organ (pancreas) and species (cat) before, which is also mentioned in the manuscript ('6 cases of primary pancreatic HSA described in the literature'). Moreover, hemangiosarcomas may occur anywhere as there are blood vessels throughout the whole body. Despite being a rare disease, hemangiosarcomas may therefore unsurprisingly affect the pancreas, too.

According to the autor guidelines of Veterinary Sciences, 'case reports usually describe new or uncommon conditions that serve to enhance medical care or highlight diagnostic approaches'. Concerning this, I do not see enough value for publication in this journal but I encourage you to publish it elsewhere.

The language style should be improved in some phrases and there are some spelling mistakes, too.

Author Response

Dear Reviewer,

Thank you very much for reviewing this case report.

Even though it is not the first case report of pancreatic hemangiosarcoma in cats, as mentioned in the text, it is the first report describing the complete pathological findings. Other cases similar to this were included only in epidemiological studies. We also performed a complete immunohistochemical evaluation and a comparative assessment of 3 anti-CD31 antibodies.

Language style and spelling mistakes were checked and corrected.

Reviewer 3 Report

This  report aims to present the pathological findings of a case of primary pancreatic hemangiosarcoma in a cat with a history of sudden death. A neoplastic mass within the body of the pancreas was detected, along with multiple metastases within the omentum and liver. The sudden death was caused by the spontaneous rupture of the pancreatic neoplasm, which produced severe hemoperitoneum and anemia.

In general, in well-differentiated vascular neoplasms diagnosis is straightforward and seldom requires immunohistochemistry. However, hemangiosarcomas may form solid sheets of spindle neoplastic cells without formation of vascular spaces, and poorly differentiated hemangiosarcomas may lack vascular channel formation altogether and morphologically mimic nonvascular spindle cell sarcomas. Moreover,  It has also been featured an epithelioid phenotype of hemangiosarcoma that could further complicates morphological diagnosis. For these reasons, immunohistochemistry is commonly used to diagnose vascular neoplasms in domestic species.

The positive immunostaining for factor VIII-related antigen is useful in differentiating poorly differentiated haemangiosarcomas  from spindle cell sarcomas or other potentially confusing non-endothelial neoplasms, such as fibrous histiocytomas.

Even if it must be remembered that negative staining for  factor VIII- related antigen does not definitely exclude haemangiosarcoma, I think that the results obtained are not sufficient to confirm the diagnosis of hemangiosarcoma.

Author Response

Dear Reviewer,

Thank you for you reviewing this case report.

The histological aspects observed in this case were mixed features of capillary, cavernous, and epithelioid, with the latter being observed multifocal, and the solid pattern not being identified, which could furthermore complicate the final diagnosis. However, we requested additional immunohistochemical evaluation. Von Willebrand factor showed no reactivity with the species as it was later added into the text.

According to Meuten "Traditionally, factor VIII immunopositivity has been considered diagnostic of hemangiosarcoma. Unfortunately, experience has shown that some hemangiosarcomas will not stain with this antibody and that some tumors with the histological appearance of lymphan- giomas or lymphangiosarcomas will stain for factor VIII. The use of CD31 (PECAM) alone or in concert with factor VIII has been shown to be more specific, and may be necessary in cases of epithelioid hemangiosarcoma"

The histological features along with the CD31 staining were considered sufficient for the final diagnosis of HSA.

Round 2

Reviewer 2 Report

Dear authors,

thank you for your reply and for the improvement of your manuscript.

I agree with your remarks and had already realized these features during reviewing the manuscript for the first time. However, I have to state that neither macroscopic nor histologic morphology of pancreatic hemangiosarcoma generally differs from that of non-pancreatic HAS. I therefore do not see enough value for publication in 'Veterinary Sciences' but I nevertheless still encourage you to publish it in a journal that supports single case reports.

Author Response

Dear reviewer,

Thank you for your comment.

The authors' opinion is that this case report can increase awareness of this uncommon location of hemangiosarcoma in cats. Both pancreatic tumors and hemangiosarcoma (regardless of location) are not that often in this species.  Even though it can arise in any location of the body, some locations are more common in both dogs and cats.

Moreover, before submitting the manuscript, we sent the abstract to the editor of the special issue "New Insights into Pancreatic Diseases in Animals" to see if it is suitable and we were encouraged to send the full manuscript.

Reviewer 3 Report

You have certainly improved the quality of the manuscript but the part relating to the clones used must be refined because it does not appear that they are validated for the feline specie. I would recommend looking for articles that have already been published to support your use.

Author Response

Dear reviewer,

Thank you for your comments.

The discussions related to the immunohistochemical evaluation have been changed according to your suggestion.